# AmiR-P³: An AI-based microRNA prediction pipeline in plants

**Sobhan Ataei**[1]*, **Jafar Ahmadi**[1]*, **Sayed-Amir Marashi**[2], **Ilia Abolhasani**[3]

1 Department of Genetics and Plant Breeding, Imam Khomeini International University, Qazvin, Iran,
2 Department of Biotechnology, College of Science, University of Tehran, Tehran, Iran, 3 Department of Computer Engineering, Sharif University of Technology, Tehran, Iran

* sobhanataei@ut.ac.ir (SA); j.ahmadi@eng.ikiu.ac.ir (JA)

## Abstract

### Background

MicroRNAs (miRNAs) are small noncoding RNAs that play important post-transcriptional regulatory roles in animals and plants. Despite the importance of plant miRNAs, the inherent complexity of miRNA biogenesis in plants hampers the application of standard miRNA prediction tools, which are often optimized for animal sequences. Therefore, computational approaches to predict putative miRNAs (merely) from genomic sequences, regardless of their expression levels or tissue specificity, are of great interest.

### Results

Here, we present AmiR-P³, a novel ab initio plant miRNA prediction pipeline that leverages the strengths of various utilities for its key computational steps. Users can readily adjust the prediction criteria based on the state-of-the-art biological knowledge of plant miRNA properties. The pipeline starts with finding the potential homologs of the known plant miRNAs in the input sequence(s) and ensures that they do not overlap with protein-coding regions. Then, by computing the secondary structure of the presumed RNA sequence based on the minimum free energy, a deep learning classification model is employed to predict potential pre-miRNA structures. Finally, a set of criteria is used to select the most likely miRNAs from the set of predicted miRNAs. We show that our method yields acceptable predictions in a variety of plant species.

### Conclusion

AmiR-P³ does not (necessarily) require sequencing reads and/or assembled reference genomes, enabling it to identify conserved and novel putative miRNAs from any genomic or transcriptomic sequence. Therefore, AmiR-P³ is suitable for miRNA prediction even in less-studied plants, as it does not require any prior knowledge of the miRNA repertoire of the organism. AmiR-P³ is provided as a docker container, which is a portable and self-contained software package that can be readily installed and run on any platform and is freely available for non-commercial use from: https://hub.docker.com/r/micrornaproject/amir-p3

The source code of AmiR-P³ is also freely available from:
https://github.com/Ilia-Abolhasani/amir-p3

**Data Availability Statement:** Source code: https://github.com/Ilia-Abolhasani/amir-p3 Docker image: https://hub.docker.com/r/micrornaproject/amir-p3 Supporting information: https://github.com/Ilia-

Abolhasani/amir-p3/tree/master/supplementary%
20information.

**Funding:** The authors received no specific funding
for this work.

**Competing interests:** The authors have declared
that no competing interests exist.

## Introduction

Three decades ago, the first microRNA (miRNA) was discovered [1]. Since then, miRNAs have been the subject of thousands of studies [2]. miRNAs play a wide range of important post-transcriptional regulatory roles in animals and plants [3–5], including fine-tuning the expression of essential genes [6,7]. In plants, miRNAs are one of the most abundant small non-coding RNAs [8]. As demonstrated in Fig 1, the exquisite machinery of miRNA biogenesis in plants involves Dicer-like (DCL) protein complexes responsible for cleaving an RNA duplex consisting of the mature miRNA and its complement, miRNA*, from an RNA molecule with an appropriate stem-loop structure. Notably, plants produce many distinct DCL/AGO-associated small RNAs other than miRNAs (*e.g.*, phasiRNAs and hc-siRNAs). Therefore, it is crucial

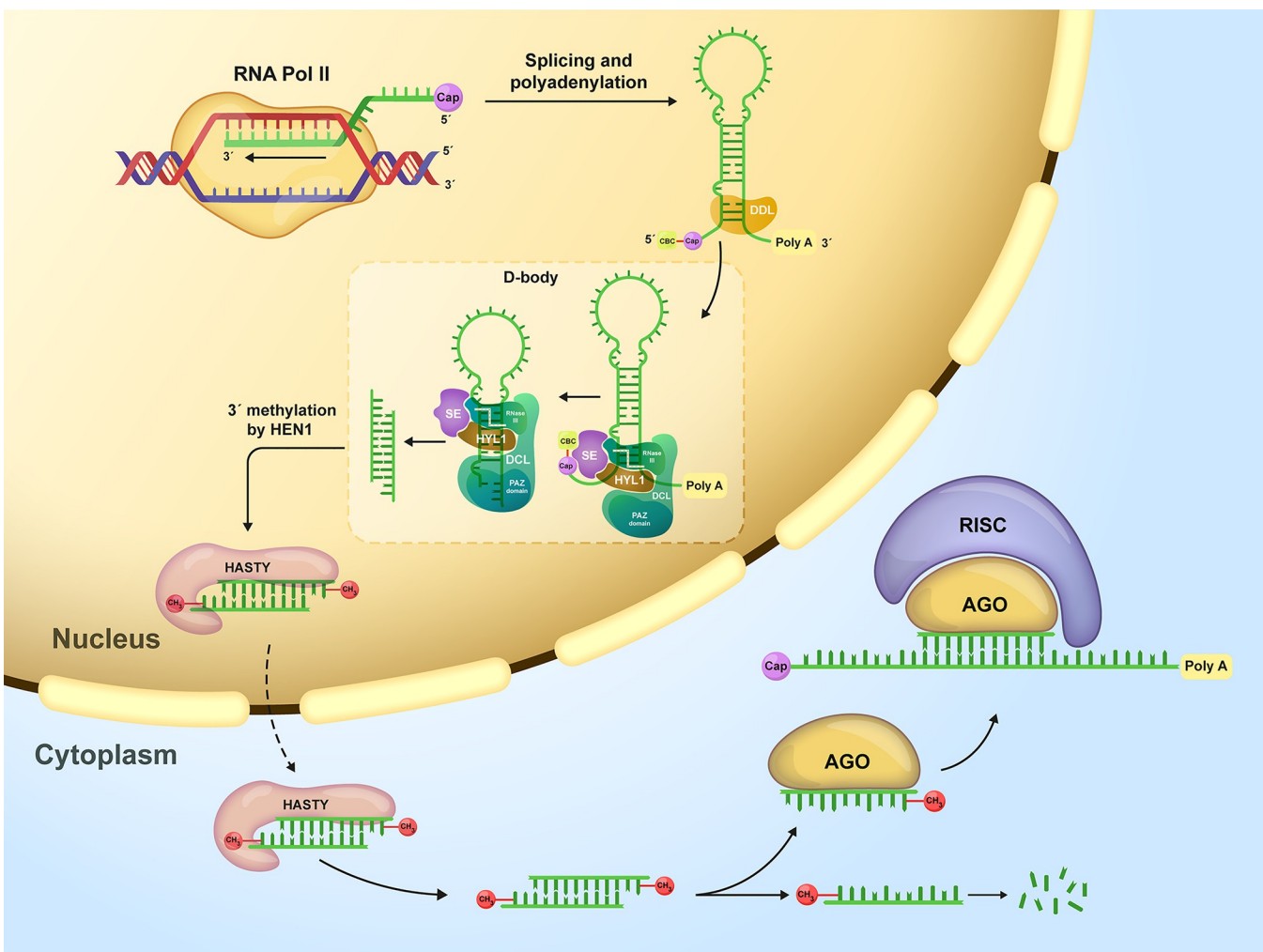

**Fig 1. miRNA biogenesis in plants.** This process begins with transcription by RNA polymerase II in the nucleus [9,10], followed by capping, polyadenylation, and splicing [11–15]. The primary transcript folds into a secondary structure, forming a double-stranded region, a terminal loop, and single-stranded flanks, referred to as primary miRNA (pri-miRNA). After stabilization by DAWDLE (DDL) proteins [16], mature miRNA is excised from pri-miRNA in nuclear bodies (D-bodies) [17,18]. Essential components include DCL1, HYL1 [19,20], SE [21,22], and CBC [23]. DCL1 cleaves pri-miRNA, generating a 3' overhang recognized by the PAZ domain. After recognizing the 3' overhang by the PAZ domain, the RNA duplex lays along the DCL1, positioning the RNase III domain about 21 bp away from the former cut site [24]. After the execution of the second cut and 3' methylation of both ends by HEN1 [25,26], the resulting RNA duplex, including guide and passenger strands, is exported to the cytoplasm by HASTY [27]. In the cytoplasm, the guide strand associates with AGO protein, forming the RNA-induced silencing complex (RISC), while the passenger strand degrades.

to consider the biogenesis pathways and structural factors that influence these routes to distinguish between different types of small RNAs.

Despite the conservedness of miRNAs [28] and their essential role in shaping plant phenotypes [29,30], our knowledge of plant miRNAs is still limited. Various approaches have been exploited to identify miRNAs in different plant species [31]. Among them, computational approaches based on high-throughput next-generation sequencing data (NGS) have shown promising results. Unlike most wet-lab methods, bioinformatics-based approaches are very economical in terms of cost, time, and labor. Moreover, such computational approaches can predict miRNAs merely based on genomic sequences, which means that the predictions are not expression-dependent and tissue-specific. This characteristic is relevant, especially in the case of low-abundance miRNAs and those with narrow spatiotemporal expression patterns.

In the field of plant microRNA prediction, several computational tools have been developed to aid researchers in identifying and characterizing these small regulatory molecules. Tools such as miRNAFinder [32], PlantMirP2 [33], mirMachine [34], PmiRDiscVali [35], miRDeep-P2 [36], SUmir [37], and miRNA Digger [38] have been instrumental in advancing our understanding of plant microRNAs. miRNAFinder utilizes machine learning algorithms to predict novel microRNAs from small RNA sequencing data. PlantMirP2 is a comprehensive tool that integrates multiple prediction algorithms to identify plant microRNAs and their targets. mirMachine employs a homology-based approach for accurate microRNA prediction. PmiRDiscVali focuses on validating predicted plant microRNAs through experimental data analysis. miRDeep-P2 is a widely used tool for discovering known and novel microRNAs from deep sequencing data. SUmir offers a user-friendly interface for predicting microRNAs and their targets in plants. Finally, miRNA Digger is a versatile tool that allows for the identification of conserved and species-specific microRNAs in plant genomes. These tools play a crucial role in expanding our knowledge of plant microRNAs and their regulatory functions, providing valuable insights into the intricate networks governing plant development and stress responses.

Although there are numerous plant miRNA prediction tools available, the majority of them exhibit limitations in terms of flexibility across three critical dimensions:

*i*) The existing miRNA prediction tools often exploit sequence conservedness and the structural and thermodynamic features of miRNAs. Homology-based miRNA prediction pipelines employ various utilities, which typically include the following steps [37,39]: (1) homology search; (2) elimination of protein-coding sequences; (3) RNA secondary structure prediction; and finally (4) extraction of structural features. Afterward, such pipelines, use a predefined set of criteria (or, alternatively, apply a classification model) to distinguish between miRNAs and other sequences. Although the outcome obviously depends on the implemented utilities, configurations, and preset parameter values, users may have limited or no access to modify these details. For example, two commonly used RNA secondary structure prediction tools in several miRNA prediction pipelines, *i.e.*, Mfold (from the UNAfold package [40,41], which is employed in C-mii [39] and SUmir [42]), and RNAfold (from the Vienna RNA package [43], which is employed in mirMachine [34]), have several adjustable parameters that are not accessible to the user of miRNA prediction pipelines. Furthermore, these tools differ in their underlying algorithms (discussed in [40]), which can lead to dissimilar results. The choice of utilities can also influence hardware requirements and processing efficiency. For example, BLASTx (from the NCBI BLAST+ suite [44]) is a pioneering protein alignment algorithm to distinguish between coding and noncoding sequences in the miRNA prediction process. Despite its extensive system requirements for big-data management, BLASTx is widely used due to its high sensitivity. On the other hand, DIAMOND [45,46] is a fast protein alignment tool that offers superior computing performance and low system requirements, with sensitivity values close to (but not necessarily as good as) those of BLASTx.

*ii*) Not all of the prediction criteria used by existing miRNA prediction pipelines are universally accepted. For instance, different miRNA prediction tools may use different thresholds for the maximum number of total and/or consecutive mismatches that are allowed between the guide and passenger strands (*i. e.*, miRNA and miRNA*). Similarly, the maximum allowed size of internal loops and bulges, the permissibility of mismatches in DICER cut sites, the permissibility of miRNA (or miRNA*) involvement in the terminal loop, and the eligibility of branched structures in the loop-proximal or loop-distal regions (outside of the miRNA/miRNA* region) are among the criteria for accepting a sequence as a miRNA, which are not universally agreed upon. Since most of the values regarding such criteria are hard-coded in the miRNA prediction programs, it is hardly possible for non-expert users to modify such features to their preferences.

*iii*) Currently, most plant miRNA prediction pipelines (*e. g.*, miRNAFinder [32], PlantMirP2 [33], mirMachine [34], PmiRDiscVali [35], miRDeep-P2 [36], SUmir [37], miRNA Digger [38], *etc.*) require high-throughput sequencing data (see S1 Table). Such data may include a reference genome (or a set of genomic reads), sRNA sequencing data, or degradome reads.

In this article, we introduce AmiR-P³, a novel *ab initio* plant miRNA prediction pipeline that benefits from the advantages of multiple computational tools. The user of this pipeline can readily adjust the prediction criteria based on the latest research on the properties of plant miRNAs. As mentioned above, AmiR-P³ does not depend on NGS reads and assembled reference genomes, and therefore, it can identify conserved and/or novel miRNAs in any genomic sequence data. More specifically, AmiR-P³ can accept any set of genomic or transcriptomic sequences as the primary input, which makes it suitable for miRNA prediction in less-studied plants.

## Materials and methods

### Preparation and evaluation of the deep-learning classifier model

**Compiling the datasets.** In the present study, we aimed to develop a machine learning-based pipeline for miRNA detection in DNA or RNA sequences. In this context, a classifier's role is to differentiate between experimentally validated miRNAs and miRNA-resembling sequences that are not miRNAs.

To train, evaluate, and validate a classifier, appropriate datasets for both positive and negative data are required. Positive data represents confirmed instances of the target class (real miRNAs), while negative data comprises instances not belonging to the target class (miRNA-resembling sequences), and, successful classification of both confirms the classifier's readiness for identifying miRNAs in new samples. Our positive datasets were obtained by extracting validated miRNA (stem-loop) sequences from the miRBase database (release 22.1). On the other hand, negative datasets were assembled by collecting sequences with predicted pri-miRNA structures but overlapping with known coding sequences (the compiled datasets are available in S1 File).

To create the positive datasets, all available stem-loop sequences of nine important plant species were separately downloaded from the miRBase database and aligned against their corresponding genomes (Table 1) using BLASTn [47]. Due to the importance of the structural features in the loop-distal region of the miRNA hairpin (*i.e.*, about 15 bp adjacent to the miRNA/miRNA* duplex in the loop-distal region) [27,48], a genomic window including the aligned sequence and two 20-nt flanking sequences on each side was selected by BEDtools [49] for further analysis. Then, we eliminated the redundant sequences using CD-HIT [50] and chose those sequences that have no overlap with known protein-coding sequences using DIAMOND [45,46]. In this step, DIAMOND was chosen over BLASTx, as it has comparable sensitivity, while it is reportedly much faster. To obtain rRNA and tRNA sequences, we used the

**Table 1. List of the plant species whose genomes were used for compiling the positive and negative datasets.**

| Species | GenBank assembly accession ID | Genome length |
|---|---|---|
| *Arabidopsis thaliana* (Thale cress) | GCA_000001735.2, TAIR10.1 | 119,668,634 |
| *Citrus sinensis* (Sweet orange) | GCA_000317415.1, Csi_valencia_1.0 | 327,669,411 |
| *Glycine max* (Soybean) | GCA_000004515.5, Glycine_max_v4.0 | 978,941,695 |
| *Gossypium raimondii* (Peruvian cotton) | GCA_000327365.2, Graimondii2_0v2 | 761,091,353 |
| *Medicago truncatula* (Barrel Clover) | GCA_003473485.2, MtrunA17r5.0-ANR | 430,008,013 |
| *Oryza sativa* (Asian rice) | GCA_001433935.1 IRGSP-1.0 | 374,422,835 |
| *Sorghum bicolor* (Great millet) | GCA_000003195.3_Sorghum_bicolor_NCBIv3 | 709,344,700 |
| *Triticum aestivum* (Bread wheat) | GCA_018294505.1 IWGSC CS RefSeq v2.1 | 14,566,954,962 |
| *Zea mays* (Maize) | GCA_902167145.1 Zm-B73-REFERENCE-NAM-5.0 | 2,182,786,008 |

Rfam database (release 14.9) [51]. Then, each of the sequences in the positive dataset was aligned against all tRNAs and rRNAs of the Rfam-derived dataset to ensure that the positive sequences are not rRNA or tRNA. Consequently, for each sequence in each of the compiled datasets, all of the optimal secondary structure(s) of its presumed RNA transcript were predicted by Mfold [41], and their structural features were extracted by CTAnalyzer [52]. It should be noted that Mfold often predicts multiple optimal structures for a single RNA sequence. So even after removing structures that do not match the plant microRNA characteristics, there may still be multiple valid structures for each pre-miRNA sequence. It is also important to note that due to either the addition of 20-nt flanks or the limitations of secondary structure prediction tools that commonly predict optimal and suboptimal RNA secondary structures, not all of the predicted secondary structures for a pre-miRNA sequence are accurate, and this can lead to uncharacteristic predicted secondary structures that do not meet the criteria for plant microRNAs. An RNA structure is said to be unacceptable for being a pri-miRNA if (*i*) the hit region is not involved in a double-stranded stem; or (*ii*) it has only a few residues in complementarity; or (*iii*) it lacks a continuous complementary region; or (*iv*) it contains inner branches; or (*v*) it is complementary to a branched region; or (*vi*) is not located entirely on the same side of the duplex (see S2 File). Consequently, after the elimination of unacceptable structures, nine positive feature datasets were obtained. Fig 2 summarizes the steps involved in compiling a positive dataset for a specific plant species (*X*).

Compiling a negative dataset involves several challenges:

*i)* All of the sequences in such a dataset should be extracted from the same plant genome sequence;

*ii)* Every such sequence, as well as the predicted secondary structure of its presumed transcript, should be "similar" to those in the positive dataset;

*iii)* Every such sequence should be chosen such that one can prove it is not a miRNA, despite the similarity of its sequence and structure to those of the genuine miRNAs; and

*iv)* The negative dataset should have the same number of sequences and the same length distribution as the corresponding positive dataset.

To create the negative (decoy) datasets, we extracted genomic windows from the genome of each of the species with a minimal similarity to real miRNAs, as explained blow:

For each of the nine plants (Table 1), mature miRNA sequences were individually downloaded from miRBase and aligned against the plant genome using BLASTn. For each pair of the aligned sequences, the level of dissimilarity (*LD*) is computed as:

$$LD = Q - Q_a + G + M$$

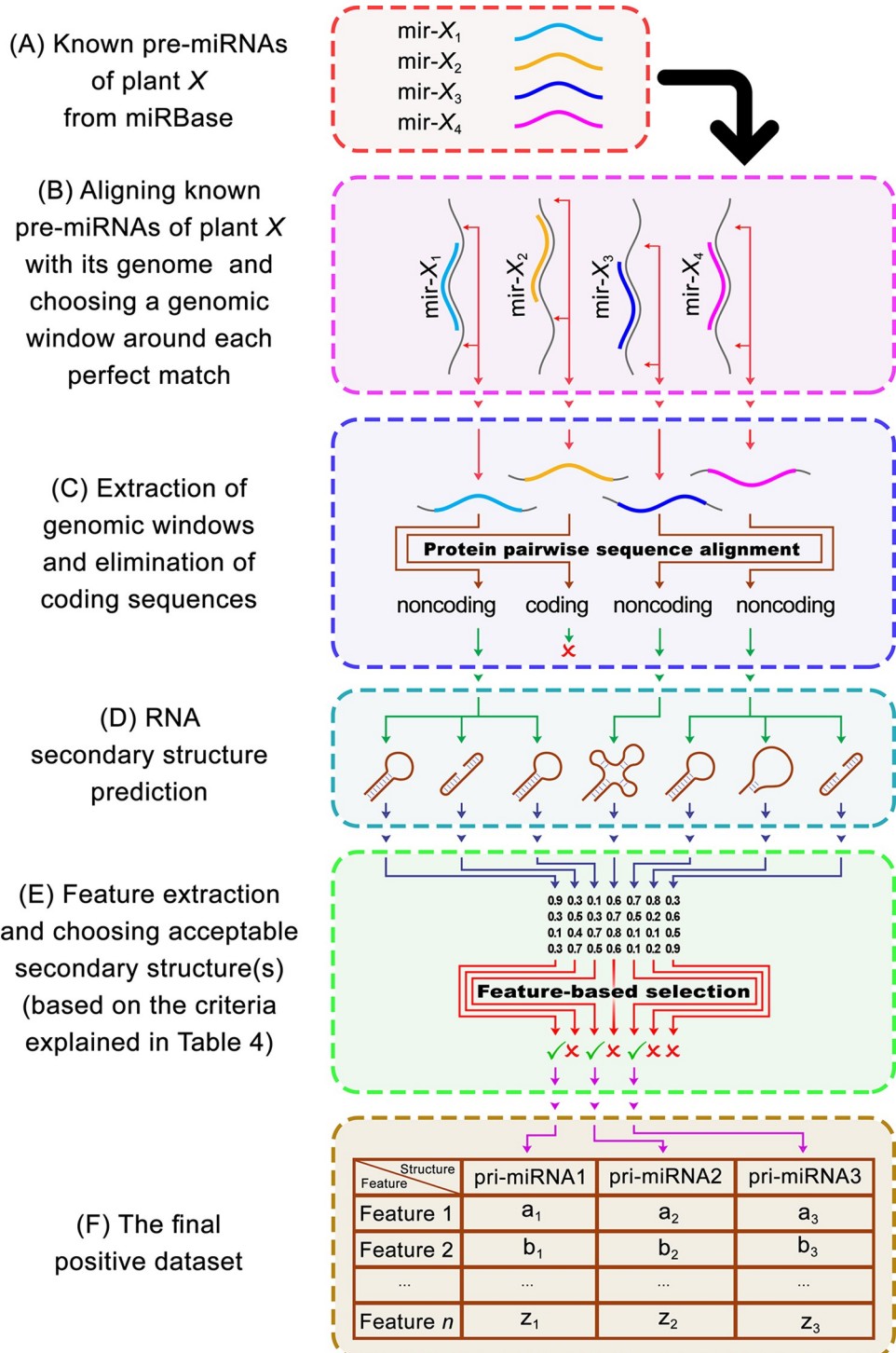

**Fig 2. Schematic representation of the procedure used for compiling each positive feature miRNA dataset.** To compile a positive dataset for a certain plant species (namely, plant *X* in this illustration), (A) all of its known pre-miRNA sequences were downloaded from miRBase, and (B) aligned with the genome of the plant species. After choosing a genomic window around each perfect match (no mismatches and gaps are accepted), (C) the extracted sequences were aligned with the GenBank nonredundant (NR) protein database and Ffam tRNA/rRNA databases, and (D) the secondary structures for sequences that have no overlap with known protein-coding sequences or any tRNAs or rRNAs were predicted. Finally, (E) after feature extraction by CTAnalyzer and the elimination of unacceptable structures (as explained in details in S2 File, structures in which the hit region is not involved in a double-stranded

stem, has only a few residues in complementarity, lacks a continuous complementary region, contains inner branches, is complementary to a branched region, or is not located entirely on the same side of the duplex), (F) the positive feature dataset for plant *X* was compiled.

Where $Q$ is the query length, $Q_a$ is the alignment length, while $G$ and $M$ are the total numbers of gaps and mismatches in the alignment, respectively. It is possible for alignments with LD<5 to be functional miRNAs [53] and alignments with LD>6 often result in E-values greater than 0.001, which is considered unacceptable and could not be considered as "significant". Therefore, only sequences with significant alignments (*E*-value ≤ 0.001) with 5≤*LD*≤6 were selected as potential hits. Then, each hit was extended from both sides to a randomly chosen length to achieve a pri-miRNA-resembling sequence. The length of each sequence was chosen such that the length distribution of the negative dataset was similar to that of the corresponding positive dataset. Then, each sequence was aligned against the GenBank nonredundant protein database (NR) using DIAMOND, to only select those sequences that overlap sequences that are translated to proteins. It is important to notice that the NR database consists of CDSs, including only the exons of eukaryotic genes. Therefore, sequences in the NR database are not expected to overlap miRNAs. Consequently, it is logical to assume that if a sequence has a significant overlap with the NR database, then there is a good chance that it is not a functional miRNA. In the next step, the secondary structure of each presumed RNA sequence was predicted by Mfold. Then, the feature extraction and the elimination of disqualified structures (based on the six rules mentioned in S2 File) were done. Finally, to create equal-sized real and decoy datasets, we randomly selected disjoint subsets of the remaining sequences as decoys (*i.e.*, non-miRNA). Each negative dataset included the same number of sequences as its positive counterpart. Fig 3 demonstrates the steps toward compiling a negative dataset for a certain plant species (namely, plant *X*). The characteristics of the compiled positive and negative datasets in this work are shown in Table 2.

In order to examine the potential bias resulting from the occurrence of nucleotide patterns in the flanking sequences of genuine and decoy pre-miRNAs within the positive and negative datasets, we employed WebLogo (version 3.7.12) [54] to create sequence logos.

## Model training

We developed a neural network with seven SELU dense layers, dropout regularization, and batch normalization to mitigate overfitting. The final layer of the model was a softmax layer that provided probability scores for each class, enabling the deep learning model to classify RNA sequences as miRNA or decoys, accurately. It should be noted that nine independently generated miRNA datasets, each representing an individual plant species, were used in this study (Table 1). The implemented approach involves training on a collection of eight distinct plant datasets, and testing the model on the remaining dataset. Notably, these datasets exhibit no interdependencies among them, ensuring a diverse range of feature distributions. This deliberate design choice was made to prevent the model from becoming species-specific, which could hinder its generalizability. Further details on the neural network model are presented in S4 File.

To evaluate our model, we performed a cross-validation analysis, in which miRNAs and decoy RNA sequences of eight plants were used to train the model, and then, to predict the miRNAs and decoys of a ninth plant. Selection of the training and testing sets from different species datasets mitigated the risk of the model merely memorizing species-specific traits, thereby promoting its capacity to learn broader patterns and features that transcend species

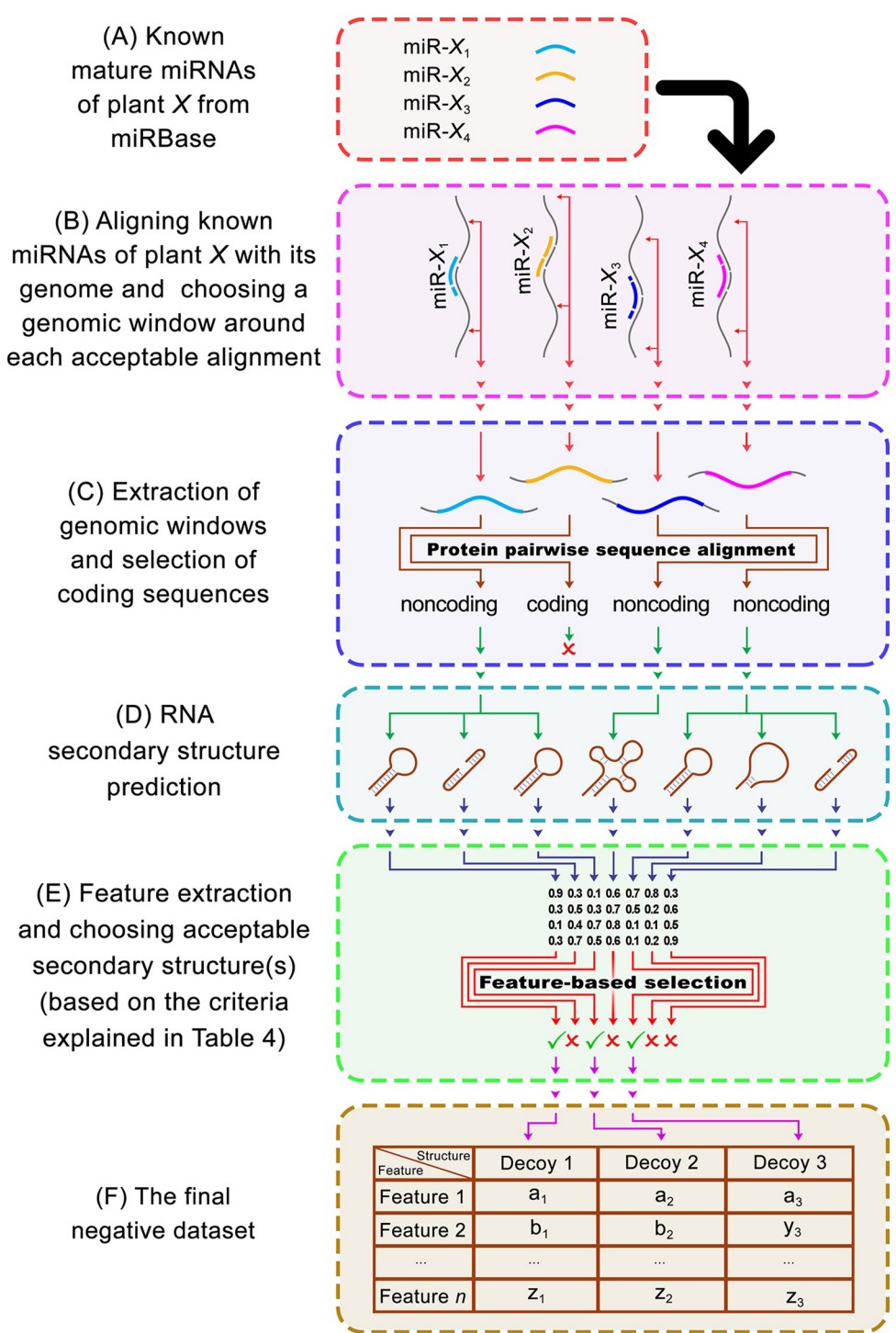

**Fig 3. Schematic representation of the procedure used for compiling each negative (decoy) dataset.** To compile a negative dataset for a certain plant species (namely, plant X), (A) all of its known mature miRNA sequences were downloaded from miRBase, and (B) aligned with the genome of the plant species. After choosing a genomic window around each acceptable alignment (E-value $\leq 0.001$ and $5 \leq LD \leq 6$), (C) the extracted sequences were aligned with the GenBank nonredundant protein database, and (D) the secondary structure for sequences that overlapped with known protein-coding sequences was predicted. (E) After feature extraction by CTAnalyzer and the elimination of disqualified structures, at the end, (F) the negative feature dataset for plant X was compiled.

**Table 2. Number of sequences included in each of the positive and negative datasets in the present work.**

| | Positive data | | | Negative data | | |
|---|---|---|---|---|---|---|
| | Pre-miRNAs in miRBase | Pre-miRNAs that overlap with no CDS | Qualified predicted structures of the accepted pre-miRNAs (the final dataset) | Extracted coding sequences with acceptable similarity to known miRNAs | Qualified predicted secondary structures of the extracted sequences | Randomly selected decoy structures in the final dataset |
| *A. thaliana* | 413 | 287 | 847 | 4550 | 5464 | 847 |
| *C. sinensis* | 234 | 185 | 540 | 743 | 914 | 540 |
| *G. max* | 773 | 428 | 1115 | 5687 | 5008 | 1115 |
| *G. raimondii* | 295 | 235 | 822 | 6650 | 10576 | 822 |
| *M. truncatula* | 713 | 434 | 971 | 5715 | 5613 | 971 |
| *O. sativa* | 690 | 471 | 1419 | 9005 | 11556 | 1419 |
| *S. bicolor* | 235 | 172 | 528 | 4099 | 3841 | 528 |
| *T. aestivum* | 170 | 151 | 421 | 11963 | 10363 | 421 |
| *Z. mays* | 308 | 204 | 625 | 1755 | 1384 | 625 |

boundaries. This analysis ensures complete independence between the test and train datasets and verifies the model's generalization ability across species.

## Components of the AmiR-P$^3$

**In-house and publicly-available tools.** In the present work, we developed AmiR-P$^3$ as a pipeline for *ab initio* identification of the conserved (and novel) plant miRNAs. The overall workflow of the pipeline is shown in Fig 4. Briefly, AmiR-P$^3$ starts with a miRNA sequence and employs several tools (Table 3) for homology search, choosing genomic windows including the potential miRNA homolog and two flanking sequences from the input sequence, redundancy removal, protein alignment, ssRNA secondary structure prediction, and feature extraction, respectively. In-house codes are also used, when necessary, for handling and analyzing the RNA sequences and their secondary structures. After the initial selection of candidate RNA structures, AmiR-P$^3$ employs the above-mentioned pre-trained classification model to distinguish genuine miRNAs from miRNA-resembling structures. At last, to fine-tune the outcome, based on the generally accepted plant miRNA identification criteria described by Axtel and Meyers [55] (see Table 4), a rule-based method is employed to verify the validity of the predicted miRNA precursors and mature miRNAs.

In addition to the above-mentioned criteria for the total number of acceptable mismatches in a miRNA/miRNA* duplex, the user can define new restrictions to further eliminate duplexes that contain any mismatches in specific positions, *e. g.*, positions 9–11.

**Details on the AmiR-P$^3$ pipeline.** In the following, we present an overview of the Amir-P$^3$ pipeline. Further technical details of the pipeline are discussed in S5 File.

At the beginning, AmiR-P$^3$ requires two datasets: a reference set of miRNAs, *e.g.*, the Viridiplantae miRNA sequences from miRBase, which can be used as queries for homology search; and a comprehensive protein dataset, *e.g.*, the NCBI-NR dataset, for sequence alignment, to distinguish protein-coding and noncoding sequences.

The input of the AmiR-P$^3$ pipeline should contain genomic or transcriptomic sequences in (multi-)FASTA format. The pipeline checks the input file format, and then, performs a BLASTn search for each sequence in the mature miRNA dataset against the input sequence(s). Consequently, around each BLASTn hit, the pipeline selects a wide genomic window, and

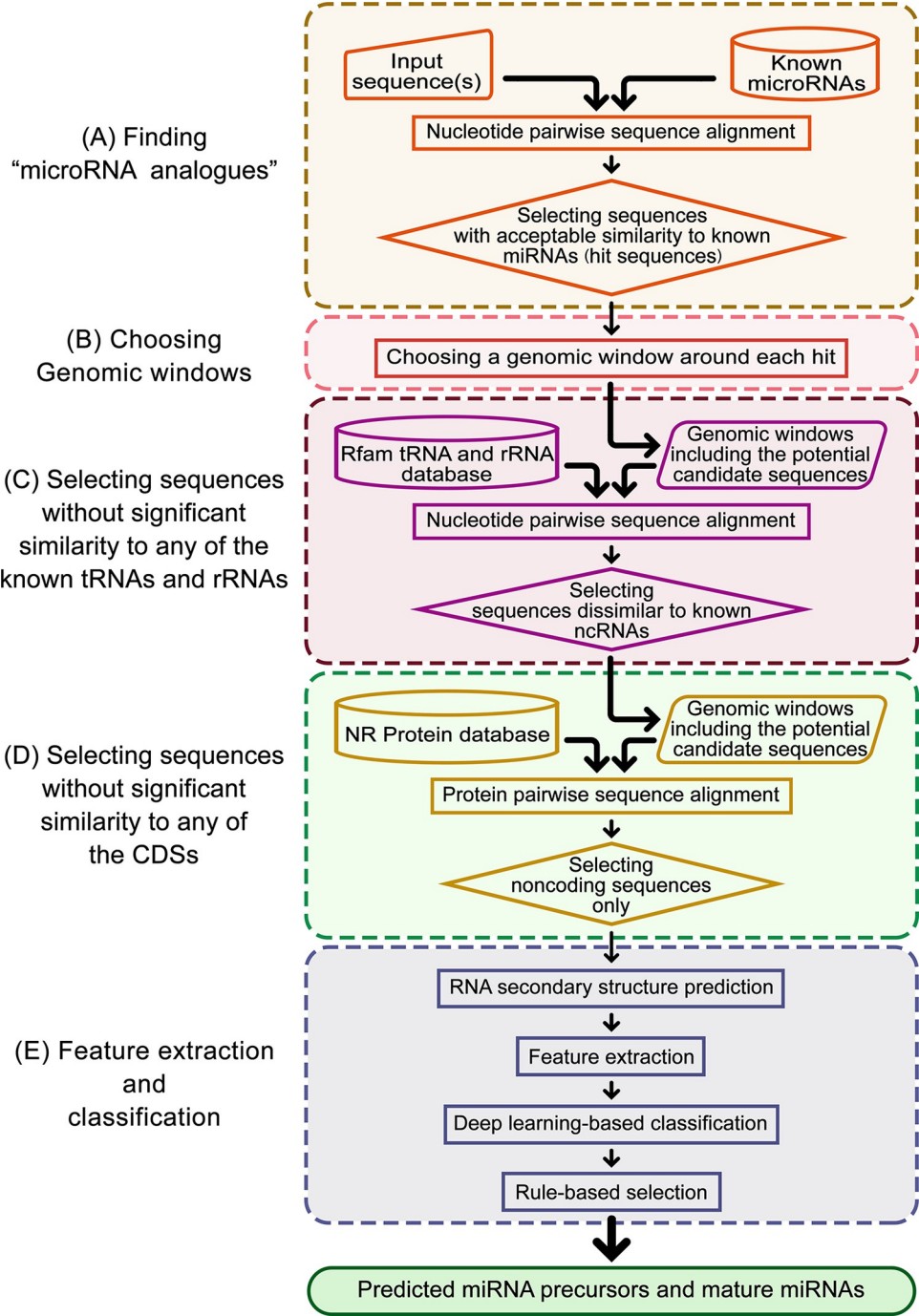

**Fig 4. Schematic representation of the AmiR-P³ pipeline.** The whole procedure includes four steps: (A) aligning the known miRNAs with the input sequence(s) (e.g., the plant genome) and selecting acceptable alignments; (B) extracting a genomic window around each hit; (C) pairwise alignment of the aforementioned sequences with the NR protein database, and selecting sequences without significant similarity to any known CDSs; and (D) RNA secondary structure prediction, feature extraction, deep learning-based classification, and rule-based selection of the pre-miRNAs. At the end of this procedure, the precursor and the mature miRNA sequences, their predicted secondary structure(s), and the extracted features of the predicted miRNA(s) are reported in the output.

**Table 3. Publicly available tools included in AmiR-P³ pipeline.**

| Application/purpose | Tool/Package |
|---|---|
| Homology search | BLASTn / NCBI BLAST+ [47] |
| Choosing a genomic window including the potential miRNA homolog and two flanking sequences | getFASTA / BEDtools [49] |
| Redundancy removal | cd-hit-est / CD-HIT suite [50] |
| Protein alignment | BLASTx / NCBI BLAST+ [47] |
|  | DIAMOND [45,46] |
| RNA secondary structure prediction | Mfold / UNAFold [41] |
|  | RNAfold / Vienna RNA package [43] |
|  | CONTRAfold [56] |
|  | MXFold2 [57] |
| Calculating the free energy of the predicted secondary structures | RNAeval / Vienna RNA package [43] |
| Extracting the features of the predicted secondary structure. | CTAnalyzer [52] |

then aligns it against the sequences in the comprehensive protein dataset (NR). This step is to ensure that each hit is not part of a coding sequence. It, then, aligns all of the remaining sequences with each sequence of the Rfam-derived dataset to remove any possible rRNA and tRNA. Finally, the secondary structure of the presumed RNA sequence is predicted.

In the next step, CTAnalyzer is employed to extract the sequence and structural features (as summarized in Table 5 and explained in detail in S2 File) and to locate the position of each presumed pre-miRNA on each of the predicted RNA structures. Additionally, CTAnalyzer is used to filter out structures that lack the basic properties of a primary miRNA (i.e., an uninterrupted stem of near-perfect complementarity between the hit region and its complementary strand; no secondary branches or large internal loops in the suggested miR/miR* region; and a simple terminal loop including at most 3 terminal structures). Then, the remaining structures are further inspected to distinguish genuine miRNA structures from miRNA-mimicking ones. To this end, three steps are taken: (1) further inspecting the remaining structures based on more precise properties (such as the size, position, and distribution of mismatches, bulges, and internal loops in the secondary structure, the GC content, and the free energy of the predicted pre-miRNA structure, etc.); (2) employing classification models, including the integrated deep learning classification model in AmiR-P3; and (3) a final rule-based examination of the qualified structures based on the miRNA sequence and structural characteristics (Table 4). AmiR-P³ is implemented in Python in the Linux/Unix environment. For simplicity, we

**Table 4. The five generally-accepted criteria for plant miRNA identification, described by Axtel and Meyers (2018) [55].**

| Criterion | Description |
|---|---|
| Typical overhangs | A miRNA/miRNA* duplex should have two-nucleotide 3' overhangs. |
| No secondary stems | No secondary stems can exist in a miRNA/miRNA* duplex. |
| A limited number of nucleotides in mismatches, internal loops, or bulges | At most six mismatched positions in the miRNA/miRNA* duplex can exist, and at most three of them can be in asymmetric bulges. |
| Pre-miRNA length limit | The length of the pre-miRNA in plants should not exceed 300 nucleotides. |
| Mature miRNA length constraints | The length of the mature miRNA can be between 20 and 24 nucleotides. |

**Table 5. A summary of sequence and structure features extracted by CTAnalyzer.**

| Type of feature | Number of features | Descriptions |
|---|---|---|
| Primary features for the secondary structures | 7 | Basic information about the acceptability of the predicted RNA secondary structure to be considered as a primary miRNA, Information about the percentage of complementarity in the hit region, and the type of the predicted miRNA (5p or 3p). |
| Reference query information | 4 | Primary information about the validated miRNA(s) which are employed as templates to identify the predicted miRNA. |
| Genomic positions | 5 | Information about the position of the miRNA genes (pre-miRNA and miRNA) on the genome and the position of the mature miRNA on the predicted pre-miRNA. |
| Nucleotide composition | 4 | Including the tetranucleotide Frequencies, GC content of the whole extracted sequence and various segments of the predicted RNA secondary structure (*i.e.*, the main branch of the predicted structure, the pri-miRNA, and the mature miRNA sequence). |
| Energy features | 8 | The free energy, MFEI (minimum free energy index), and AMFE (adjusted minimum free energy) of the main branch and the predicted pre-miRNA, along with the free energy and the MFEI for the whole predicted RNA structure. |
| Linear sequence features | 23 | Linear position (*i. e.*, start point, and end point) of various properties of the predicted secondary structure on the extracted sequence. |
| Composition of nucleotides in pairs | 36 | The state of nucleotide pairs (types of matches or mismatches) in crucial positions of the predicted secondary structure (*i.e.*, the segments adjacent to the DICER cut sites and the seed region) |
| Major features of the secondary structures | 83 | Detail of the properties of the predicted secondary structure (*e. g.*, the size and the position of all the mismatches, asymmetric bulges, internal and apical loops, secondary stems, *etc.*). |

provide AmiR-P[3] as a Docker container (freely available from https://hub.docker.com/r/micrornaproject/amir-p3), allowing the users to install the pipeline in a single step without the need to pre-install any other utilities. The source code of AmiR-P[3] and all the other data necessary to make the research reproducible (including the training datasets) are freely available from https://github.com/Ilia-Abolhasani/amir-p3. Technical details of the AmiR-P[3] pipeline are discussed in S5 File.

## Evaluation of the classification model

To evaluate the performance of the classification model, we performed a "versatility analysis", and additionally, ten-fold cross-validation analyses for the nine plants (see below).

The versatility analysis was done as follows:

1. For each plant species, the corresponding positive-negative dataset pair was left out. The remaining eight dataset pairs were used to train the machine-learning model.

2. Step 1 was repeated for each of the eight remaining plant species.

The performed ten-fold cross-validations for each plant (*i. e.*, for each of the nine compiled positive-negative dataset pairs) were as follows:

1. Each positive dataset was evenly split, at random, into 10 subsets. Each negative dataset was also partitioned accordingly. Ten positive-negative subset pairs were obtained.

2. One of the positive-negative subset pairs was set aside, while the remaining nine pairs were utilized as the training set in the machine learning process. The performance of the model was then assessed using the set-aside subset pair.

3. Step 2 was repeated for the nine remaining subset pairs.

4. Finally, the average performance scores across all validation sets were calculated.

For both of the evaluation methods, seven important correctness metrics, *i.e.*, accuracy, sensitivity, specificity, precision, F1 score, MCC (Matthew's correlation coefficient), and AUC (Area Under the ROC Curve), were calculated to verify the validity of the method. To assess the overall predictive power of the deep learning model across all species, we merged the positive data across species to obtain a dataset of all of the miRNAs in the nine plants. A similar procedure was used to obtain the negative dataset. Then, we performed a ten-fold cross-validation. The indicated correctness metrics were then computed to estimate the overall performance of the model.

## Comparing AmiR-P$^3$ with MiRFinder and PlantMirP2

MiRFinder [58] is a pioneer miRNA prediction tool that is accurate and efficient in predicting pre-miRNA sequences from genomic DNA. MiRFinder works in two steps. First, it scans the genome for potential pre-miRNA hairpin sequences. Once potential pre-miRNA hairpin sequences have been identified, it uses a support vector machine (SVM) classifier to classify them as either miRNA or non-miRNA hairpins. The SVM classifier uses some structural features to distinguish between miRNAs and non-miRNA hairpin sequences. PlantMirP2 is another plant miRNA prediction tool that uses a set of energy-based features to train an SVM model for detecting miRNAs with high accuracy. Despite differences in feature extraction and miRNA prediction strategies, these two miRNA prediction tools (with a few minor modifications) can be evaluated using the benchmark datasets established for evaluating AmiR-P$^3$.

To evaluate the AmiR-P$^3$ pipeline, in each iteration, the positive and negative datasets of eight plants were used to train the pipeline, while the correctness of the predictions was evaluated on the remaining plant. For MiRFinder and PlantMirP2, predictions were made by software tools with their default parameter settings. Based on these predictions, the correctness measures were computed individually for each of the nine plants.

## Example: The case of *Azadirachta indica*

To show the capability of AmiR-P$^3$ for discovering miRNAs of understudied plants, we chose *Azadirachta indica* as a case. *A. indica* is a valuable plant due to its ability to produce versatile secondary metabolites [59]. At the same time, the availability of its complete genomic sequence makes it an appropriate species for our analysis. To the best of our knowledge, no *A. indica* miRNA is currently available from miRBase or other miRNA repositories. To identify the putative miRNAs of *A. indica*, its latest genome sequence (GenBank assembly accession ID: GCA_022749755.1) was downloaded from the NCBI database and introduced as the input to the pipeline to perform a genome-wide miRNA prediction. For the homology search, we set the maximum acceptable *LD* value to 3 [60] and, for each hit, a genomic window that included the hit and two 200 nt flanking sequences on each side was chosen [61,62] for RNA secondary structure prediction.

## Results and discussion

### Checking the compiled datasets for sequence bias

To assess bias caused by nucleotide patterns in flanking sequences of real and decoy pre-miRNAs in both positive and negative datasets, we generate sequence logos using WebLogo. The generated logos, available in S3 File, illustrate the 20 nucleotides both upstream and downstream of each genuine and decoy pre-miRNA in both the positive and negative datasets, demonstrate that there is no distinct nucleotide conservation evident in either of the datasets.

**Table 6. Results of the ten-fold cross-validation analysis for predicting miRNAs in the nine plants.**

|  | Accuracy | Precision | Sensitivity | Specificity | F1 score | MCC | AUC |
|---|---|---|---|---|---|---|---|
| *A. thaliana* | 0.9787±0.0119 | **0.9759±0.0183** | **0.9824±0.0149** | 0.9747±0.0204 | **0.9790±0.0112** | 0.9574±0.0237 | **0.9966±0.0042** |
| *C. sinensis* | **0.9796±0.0150** | 0.9754±0.0204 | 0.9823±0.0212 | **0.9765±0.0192** | 0.9787±0.0166 | **0.9592±0.0302** | 0.9904±0.0118 |
| *G. max* | 0.9283±0.0146 | 0.9197±0.0285 | 0.9377±0.0223 | 0.9199±0.0258 | 0.9283±0.0163 | 0.8563±0.0296 | 0.9799±0.0076 |
| *G. raimondii* | 0.9428±0.0101 | 0.9502±0.0185 | 0.9346±0.0243 | 0.9495±0.0217 | 0.9420±0.0114 | 0.8859±0.0200 | 0.9816±0.0067 |
| *M. truncatula* | 0.9088±0.0179 | 0.9019±0.0326 | 0.9180±0.0283 | 0.9008±0.0339 | 0.9093±0.0182 | 0.8184±0.0350 | 0.9606±0.0195 |
| *O. sativa* | 0.9419±0.0080 | 0.9368±0.0236 | 0.9486±0.0225 | 0.9352±0.0269 | 0.9422±0.0077 | 0.8845±0.0155 | 0.9842±0.0053 |
| *S. bicolor* | 0.9385±0.0190 | 0.9124±0.0428 | 0.9704±0.0268 | 0.9088±0.0416 | 0.9396±0.0198 | 0.8799±0.0348 | 0.9873±0.0073 |
| *T. aestivum* | 0.9407±0.0273 | 0.9301±0.0363 | 0.9530±0.0351 | 0.9239±0.0499 | 0.9409±0.0279 | 0.8807±0.0547 | 0.9806±0.0157 |
| *Z. mays* | **0.9944±0.0054** | **0.9906±0.0107** | **0.9983±0.0055** | **0.9904±0.0114** | **0.9944±0.0053** | **0.9889±0.0107** | **0.9999±0.0001** |

The values are presented in "mean±std" format. For each of the correctness measures, the best two values are highlighted in the bold-underlined font.

## Model validation

**Ten-fold cross-validation.** To evaluate the classification model which is employed to identify miRNAs based on the predicted secondary structure, a ten-fold cross-validation analysis was performed separately for each pair of the nine species (Table 6). According to the calculated metrics, the best predictions are observed in the case of *Z. mays*, followed by *A. thaliana* and *C. sinensis*. Details on all of the calculated metrics are presented in S2 Table.

To assess the overall performance of the model across all species, we merged the positive data across species to obtain a dataset of all of the miRNAs in the nine plants. A similar procedure was used to obtain the negative dataset. Then, we performed a ten-fold cross-validation. The calculated correctness metrics are presented in Table 7.

**Versatility analysis.** A versatility analysis was performed to evaluate the relevance of applying our classification model for predicting miRNAs in various plant species. More precisely, in each iteration, the classification model was trained on eight plant datasets, that is, eight pairs of positive and negative datasets, and then, tested on the remaining plant dataset. Afterward, the evaluation metrics were calculated for the test dataset. Details on datasets are provided in Table 2. A Comparative plot for the values of accuracy, sensitivity, specificity, precision, F1 score, and AUC is presented in Fig 5. The ROC (receiver operating characteristic) curve for the versatility analysis is also shown in Fig 6. All these results suggest that, in most cases, our machine-learning model can be satisfactorily trained to distinguish real pre-miRNAs from decoy sequences. Therefore, we used this classifier in our pipeline to predict miRNAs.

## Comparing the predictive power of AmiR-P³, MiRFinder, and PlantMirP2

In the next step, to make a comparative evaluation of our pipeline, we used nine pairs of positive and negative datasets consisting of predicted RNA secondary structures from real pre-miRNAs and decoy sequences (S6 File) to the classification models of MiRFinder, PlantMirP2, and AmiR-P³. Then we calculated the evaluation metrics for the three programs. As presented in Fig 7, AmiR-P³ outperformed MiRFinder in accuracy, precision, sensitivity, F1 score, and MCC. It also exceeded PlantMirP2 in accuracy, precision, specificity, F1 score, and MCC.

**Table 7. Results of the ten-fold cross-validation analysis for the union of all miRNAs in the nine plants.**

|  | Accuracy | Precision | Sensitivity | Specificity | F1 score | MCC | AUC |
|---|---|---|---|---|---|---|---|
| Mean | 0.9476 | 0.945 | 0.9505 | 0.9445 | 0.9477 | 0.8952 | 0.9856 |
| Std | 0.0043 | 0.0072 | 0.0094 | 0.0086 | 0.0046 | 0.0086 | 0.0022 |

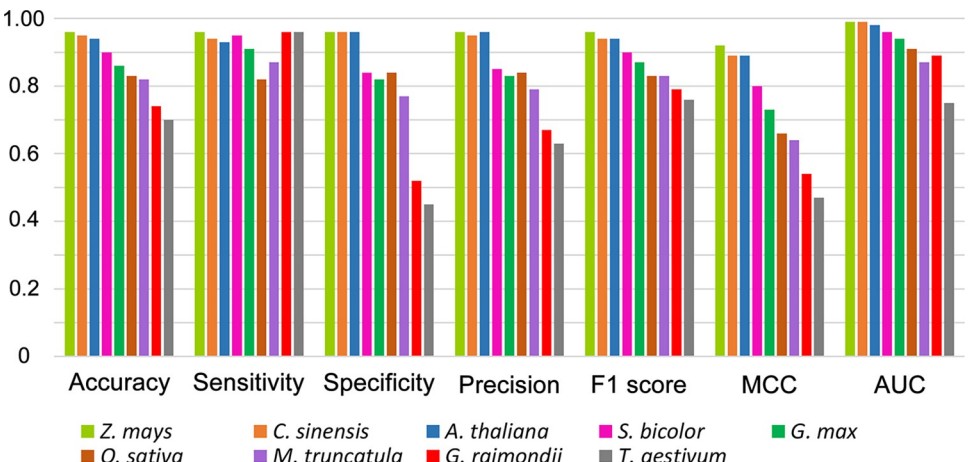

**Fig 5. The comparative plot of evaluation metrics of the versatility analysis.** The calculated values of accuracy, sensitivity, specificity, precision, F1 score, and AUC for the nine iterations of versatility analysis performed on the classification model of AmiR-P$^3$.

## Case study: miRNAs of *A. indica*

We chose *A. indica* for a case study because it is a valuable but understudied plant with a publicly available genome sequence. For the homology search, all 10414 available mature Viridiplantae miRNAs in the miRBase database (release 22.1) were downloaded. Afterward, cd-hit was applied to create a nonredundant plant miRNA dataset, resulting in 6028 unique subgroups. Then, these sequences were used as a query to find potential homologs in the *A. indica* genome. This genome-wide search resulted in 13983 hits.

When finding plant microRNAs based on homology to known miRNAs, a genomic window is chosen around each hit. Plant miRNA precursors show a broad distribution of length. The window is chosen to be large enough to encompass the entire pri-miRNA sequence. If we

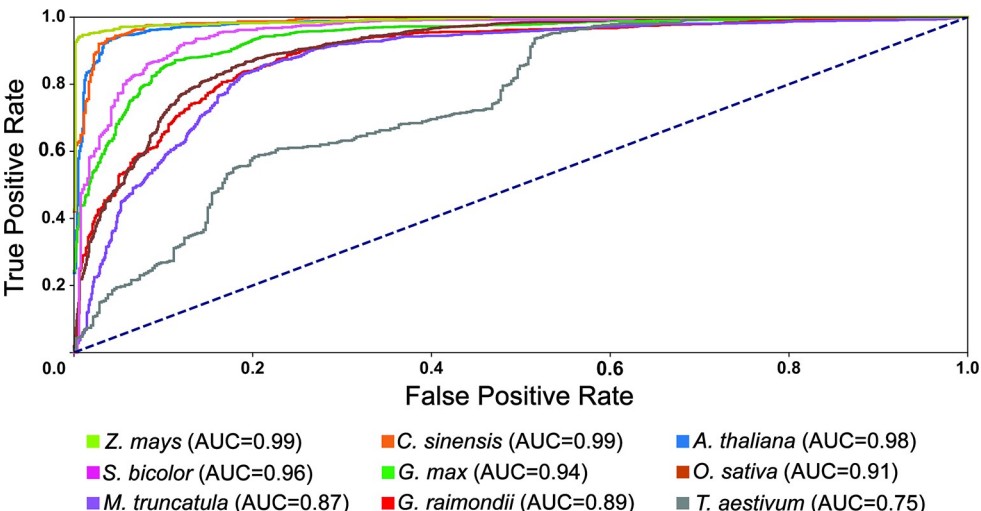

**Fig 6. ROC curves and the corresponding AUC values of the versatility analysis of the classification model of AmiR-P$^3$.** These results confirm the capability of the developed classification model to distinguish real pre-miRNAs from decoy sequences.

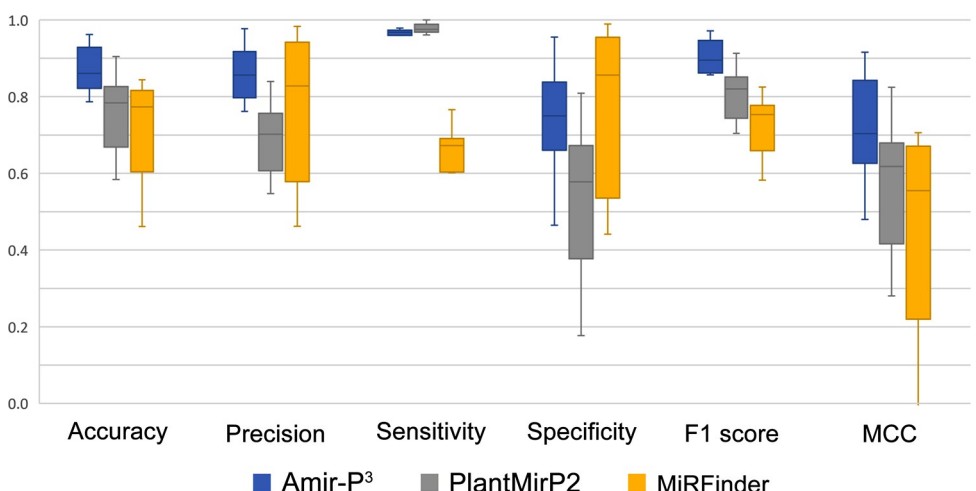

**Fig 7. Comparison of evaluation metrics for the classification models of AmiR-P³, MiRFinder and PlantMirP2.**
Results indicate that AmiR-P³ outperformed MiRFinder in accuracy, precision, sensitivity, F1 score, and MCC. It also exceeded PlantMirP2 in accuracy, precision, specificity, F1 score, and MCC.

choose a very short genomic window, we may not capture the entire pri-miRNA sequence. This can make it difficult to confirm the sequence to be a miRNA. On the other hand, using a very large genomic window can increase the false positive rate. This is because a very large window is more likely to include sequences that, when transcribed, have the characteristics of miRNA sequences and structures. Additionally, handling large genomic windows are more computationally expensive than smaller windows, especially in RNA secondary structure predictions. Suppose that the length of our hit sequence (*i. e.*, the genuine miRNA) is *n*. Our goal is to select a window that includes the hit sequence and two flanking regions around it. In different studies, various window sizes have been selected, ranging from 100+*n*+100 [63] to 250+*n*+250 [64] nucleotides. In this study, in the case of *Azadirachta indica*, we chose a 200+*n*+200 = *n*+400 nt window (including the hit and two 200 nt flanking sequences on each side) [61,62]. We chose this value such that the length of about 98% of pre-miRNAs of viridiplantae in miRBase database (release 22.1) is covered. In the next step, by choosing DIAMOND as the protein alignment tool, protein-coding sequences were found and removed from the set of potential pri-miRNAs. In the next step, assuming that each of the 9225 remaining sequences can be transcribed to a noncoding RNA, all possible secondary structures of each such RNA were predicted by Mfold. Next, the 136211 predicted structures were submitted to the CTAnalyzer for feature extraction. The primary selection criteria removed 121074 disqualified structures, leaving only 15137 structures. These structures were expected to include both real miRNA and miRNA-mimicking structures. In the next step, the trained deep learning classification model predicted 2386 structures as the positive class. Finally, based on the preset miRNA prediction criteria, 240 mature miRNA sequences from 345 pri-miRNA structures were selected as the acceptable miRNA set. Details on the predicted *A. indica* miRNAs, including reference miRNA IDs and the species in which the reference miRNA is reported, precursor and mature miRNA sequences of *A. indica*, their genomic position, and the calculated probability values are presented in S3 Table. We also highlighted those miRNAs that were recently observed experimentally [53,65,66].

Looking ahead, the elucidation of the predicted miRNAs in *A. indica* opens up a realm of exciting future prospects with significant implications. One of the foremost avenues lies in unraveling the functional significance of these identified miRNAs. Validating the predicted

target genes through experimental approaches such as RNA-seq and qPCR will provide a deeper understanding of the regulatory roles these miRNAs play in key biological processes. Moreover, these predicted miRNAs hold immense potential for biotechnological applications, particularly in the realm of metabolic engineering. Their role in modulating the synthesis of beneficial metabolites unique to *A. indica* could be harnessed to enhance the plant's medicinal and therapeutic properties. By integrating miRNA data into metabolic networks, we can gain insights into how these miRNAs influence the production of bioactive compounds. Furthermore, these miRNAs offer a platform for targeted enhancement of specific metabolites, thereby contributing to the overall medicinal value of *A. indica*. As we embark on these exciting research trajectories, the predicted miRNAs stand as promising candidates with far-reaching implications for both fundamental plant biology and the advancement of biotechnological applications.

## An overview of the available tools for predicting plant miRNAs

Numerous software tools are available for the identification of microRNAs in plants, as outlined in Table 8. Despite variations in algorithms and identification criteria, many of these tools benefit significantly from the utilization of data obtained through next-generation sequencing. This advantage is attributed to the ability to confidently identify and introduce known or novel microRNA molecules based on biological evidence, using sequences derived from RNA extraction across diverse plant tissues. In cases where the prediction tool employs additional data, such as degradome sequencing data (*e. g.*, miRNA Digger [38]) or annotation files (*e. g.*, miR-PREFeR [67]), more reliable predictions are expected. However, it is crucial to acknowledge that, despite the credibility associated with miRNA identification based on NGS,

**Table 8. An overview of the available tools for predicting plant miRNAs.**

| Software name | Method | Necessary inputs |
|---|---|---|
| miRNAFinder [32] | MLP-based classifier | • NGS reads<br>• Reference genome<br>• ncRNA sequences |
| PlantMirP2 [33] | SVM-based classifier | • Simple FASTA sequence for pre-microRNA prediction.<br>• NGS data file, genome data file, and ncRNA data file for mature microRNA prediction |
| mirMachine [34] | Rule-based prediction | • Formatted NGS reads<br>• Reference genome |
| PmiRDiscVali [35] | Based on miRDeep-P | sRNA NGS readsRNA NGS readsOptional degradome NGS reads<br>• Optional reference genome |
| miRDeep-P2 [36] | RF-based classifier | NGS formatted reads<br>• Reference genome |
| SUmir2 [37] | Rule-based prediction | • High-throughput genomic and transcriptomic sequences in FASTA format or sRNA sequencing data |
| miRNA Digger [38] | Rule-based prediction | Degradome sequencing data<br>• Reference genome |
| miRPlant [68] | Based on miRDeep* | NGS sRNA reads<br>• Reference genome |
| miPlantPreMat [69] | SVM-based classifier | • Simple FASTA sequence |
| miR-PREFeR [67] | Rule-based prediction | NGS sRNA readsReference genome<br>• Optional annotation file |
| C-mii [39] | Rule-based prediction | • Simple FASTA sequence |

this approach is not without limitations. Although nucleic acid sequencing is now more accessible and cost-effective, sequencing-based methods remain resource-intensive in terms of both cost and time. Additionally, when microRNA identification relies on RNA extraction, limitations persist in terms of restricted identification to specific tissues, developmental stages, and, more importantly, miRNA expression levels. In essence, these methods primarily target the identification of the most abundant microRNAs. Consequently, the likelihood of non-identification of microRNAs with minimal expression levels and limited occurrences in tissue samples is significant when employing RNA NGS-based approaches. The key feature that renders the utilization of AmiR-P³ valuable alongside other microRNA identification tools is its capability to predict microRNAs solely based on genomic sequences, without the need for NGS data. Consequently, AmiR-P³ will be able to identify microRNAs that might have very limited expression in cells.

As shown in details in Table 8, few plant miRNA prediction tools are capable to identify plant miRNAs using only simple FASTA sequences (*e. g.*, C-mii [39] and miPlantPreMat [69]) and the majority of well-known miRNA prediction tolls such as miRDeep-P2 [36], require several mandatory inputs such as sRNA NGS reads and reference genomes as well as optional degradome sequencing data (*e. g.*, miRNA Digger [38]) or optional annotation files (*e. g.*, miR-PREFeR [67]) and, to the best of our knowledge, none of currently available plant miRNA prediction tools use a two-stepped verification method (*i. e.*, a feature-based selection of candidate pre-miRNA structures after the AI-based classification) to identify miRNAs. So, the minimal input data requirement in AmiR-P³ makes it suitable for studying orphan species. Additionally, the two-stepped verification method is believed to yield more reliable miRNA identifications.

## Conclusion and future work

In this article, we introduced AmiR-P³, a machine learning-based (plant) miRNA prediction pipeline, which exploits multiple utilities for its key computational steps. Due to its minimal input requirements, AmiR-P³ is especially recommended for *ab initio* miRNA prediction, even in less-studied plants. AmiR-P³ is applicable to the discovery of miRNAs in a non-tissue specific and non-expression dependent manner, especially in the case of low abundance miRNAs, and those with narrow spatiotemporal expression patterns. The efficacy of AmiR-P³ is demonstrated by its notable evaluation results. The efficiency of the pipeline is expected to improve as the number of validated plant miRNAs used to train the classification model increases. Furthermore, the same procedure can be used to tailor the classifier and the pipeline for other species, including animals.

In this work, we kept all of the features, including the correlated ones, for training our deep learning model. Here, we would like to point out that the decision to drop or retain correlated features is not straightforward. In our case, the deep learning model, in contrast to linear models such as logistic regression, can handle correlated features. Furthermore, we adjusted the dropout rates, layer depths, and network architecture to optimize the model performance and avoid overfitting. While dropping correlated features can reduce multicollinearity problems and simplify the model, it can also result in losing valuable information that might be relevant for the prediction. Therefore, we decided to keep all the features in our model, as we believe that they collectively contribute to the accuracy of the classification. One may still ask whether dropping correlated features can improve the results. However, answering this question requires conducting a separate set of experiments, comparing the model performance with and without these features, which can be a topic for future work. The correlation between the features used in the training process is presented in S7 File as a heatmap.

## Supporting information

**S1 Table. An overview of the available tools for predicting plant miRNAs.** comprehensive information about some of the available plant miRNA prediction tools, including the prediction methods and necessary inputs.
(DOCX)

**S2 Table. Details on all of the calculated metrics in ten-fold cross-validations.** All the calculated values in ten-fold cross-validations.
(XLSX)

**S3 Table. Details of all *A. indica* pre-miRNAs and miRNAs predicted by AmiR-P$^3$.** Full details of the 240 predicted microRNAs in 345 precursors for *Azadirachta indica*.
(XLSX)

**S1 File. Classification model training datasets.** The compiled positive and negative datasets employed to train and test the implemented classification model.
(ZIP)

**S2 File. An introduction to CTAnalyzer.** The software tool employed for extracting features from the predicted secondary structures of confirmed miRNAs, decoy sequences and candidate pri-miRNAs.
(DOCX)

**S3 File. Sequence Logos of 20nt flanking regions in positive and negative Pre-miRNA datasets.** Sequence logo illustrations to examine the potential bias resulting from the occurrence of nucleotide patterns in the flanking sequences of genuine and decoy pre-miRNAs within the positive and negative datasets.
(DOCX)

**S4 File. Details on the implemented neural network.** Details on the developed neural network employed to classify the predicted RNA secondary structures.
(DOCX)

**S5 File. Technical details of AmiR-P$^3$ pipeline.** Details on general structure and prediction approach of AmiR-P$^3$.
(DOCX)

**S6 File. Comparative datasets.** The datasets employed to Compare AmiR-P$^3$ *vs* PlantMirP2.
(ZIP)

**S7 File. Heatmap.** The correlation between the features used in the training process presented as a heatmap. All of the supporting information is also available on GitHub at https://github.com/Ilia-Abolhasani/amir-p3/tree/master/supplementary%20information.
(DOCX)

## Acknowledgments

The authors would like to acknowledge the assistance of Blake C. Meyers (University of Missouri), Tanushree Tiwari (York University), Kaveh Kavousi (University of Tehran) and Mohammad Hossein Norouzi-Beirami (Islamic Azad University, Osku) in providing technical guidance and insightful discussions throughout this work.

## Author Contributions

**Conceptualization:** Sobhan Ataei, Jafar Ahmadi, Sayed-Amir Marashi, Ilia Abolhasani.

**Data curation:** Sobhan Ataei, Ilia Abolhasani.

**Formal analysis:** Sobhan Ataei, Sayed-Amir Marashi.

**Investigation:** Sobhan Ataei, Sayed-Amir Marashi.

**Methodology:** Sobhan Ataei, Sayed-Amir Marashi, Ilia Abolhasani.

**Project administration:** Sobhan Ataei, Jafar Ahmadi, Sayed-Amir Marashi.

**Resources:** Jafar Ahmadi.

**Software:** Ilia Abolhasani.

**Supervision:** Jafar Ahmadi, Sayed-Amir Marashi.

**Validation:** Jafar Ahmadi.

**Visualization:** Sobhan Ataei, Sayed-Amir Marashi, Ilia Abolhasani.

**Writing – original draft:** Sobhan Ataei.

**Writing – review & editing:** Jafar Ahmadi, Sayed-Amir Marashi.

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
