## [Decision Letter · Decision Letter 0]

4 Jul 2023

PONE-D-23-17993AmiR-P3: An AI-based microRNA prediction pipeline in plantsPLOS ONE

Dear Dr. Ataei,

Thank you for submitting your manuscript to PLOS ONE. After careful consideration, we feel that it has merit but does not fully meet PLOS ONE’s publication criteria as it currently stands. Therefore, we invite you to submit a revised version of the manuscript that addresses the points raised during the review process.

We look forward to receiving your revised manuscript.

Kind regards,

Vibhav Gautam

Academic Editor

PLOS ONE

Reviewers' comments:

Reviewer's Responses to Questions

**Comments to the Author**

1. Is the manuscript technically sound, and do the data support the conclusions?

Reviewer #1: Yes

Reviewer #2: Partly

Reviewer #3: Yes

2. Has the statistical analysis been performed appropriately and rigorously? 

Reviewer #1: Yes

Reviewer #2: Yes

Reviewer #3: Yes

3. Have the authors made all data underlying the findings in their manuscript fully available?

Reviewer #1: Yes

Reviewer #2: No

Reviewer #3: Yes

4. Is the manuscript presented in an intelligible fashion and written in standard English?

Reviewer #1: Yes

Reviewer #2: Yes

Reviewer #3: Yes

5. Review Comments to the Author

Reviewer #1: The manuscript entitled “AmiR-P3: An AI-based microRNA prediction pipeline in plants” by Ataei et al. have developed a machine learning-based pipeline for plant miRNA prediction tool named AmiR-P3. The AmiR-P3 is an ab initio miRNA prediction tool with a requirement of minimalistic input, and also helpful in predicting miRNAs, even in less-studied plants. The authors identified many novel miRNAs in Azadirachta indica. The work is interesting and relevant, because it will help to predict miRNAs in newly sequenced plant genomes, those having no reference genome. However, the manuscript is well written but somewhere it does not very clear about the usage of parameters/tools. Following are some concerns which need to be addressed:

1. Authors stated that, to train a classifier to identify miRNAs, 2 datasets were used, one of known miRNAs and the other of miRNA-resembling sequences that are not miRNAs. For known miRNAs authors used miRBase, but did not explain from where they collected the transcripts resembled with pri-miRNA structures having overlaps with known coding sequences. Specify and explain about the negative dataset.

2. Authors must have to explain about the significance of taking LD value between 5 and 6.

3. The authors used DIAMOND software for alignment, though they are using BLAST, explain the usage priority of DIAMOND software.

4. Whether authors checked for tRNA or rRNA folds before qualifying non-coding sequences as pre-miRNAs predicted structures for positive dataset?

5. The authors mentioned that they have allowed six mismatched positions in the miRNA/miRNA* duplex, and mainly allowed three at asymmetric bulges. But it is assumed that the mismatches at position 9-11 will lead to non-functional miRNA. Whether authors checked the mismatch positions properly?

6. Why authors compared AmiR-P3 only with PlantMirP2? Why not with other miRNA prediction tools?

Reviewer #2: Authors have presented a miRNA prediction tool . However, the following comments should be addressed before it publication.

Major comments

Mention the criteria of eliminating unacceptable in the main text.

Have you observed any pattern of nucleotides in the flanking sequences of +ve and -ve miRNAs. Present a sequence logo to show type of nucleotide conservation in the miRNA flanking sequences.

if you have taken all the validated miRNAs, then how their structures can be unfavourable?

Why authors have taken pre-miRNAs in positive dataset and pseudo miRNAs in Negative dataset.

Mention the basis of selecting the specific window size in flanking sequences and also state what would have happened if a different window size would have been taken.

Clear explanation is required for more number of qualified predicted structures of accepted miRNAs than the pre miRNAs of mirBase.

If a validated miRNA overlaps with CDS than it suggests the possibility of miRNA biogenesis from a coding region. If so, how the selection of negative miRNA dataset from coding region is justified.

Authors have used a total of 170 features. They should analyze the correlation between features and present the results. if some features are highly correlated then only uncorrelated features should be used.

A paragraph on data processing result with suitable figure/table should be presented under Results section.

Have authors compared their approach with other methods using an independent test dataset. If yes then proved details of the independent test dataset.

From the AUC-ROC, it seems the accuracy of pipeline varies from species to species. Have the authors merged all the positive data together and all the negative data together irrespective of species and performed 10 fold cross validation. If so, then provide details otherwise carry out the exercise and present the results to estimate the overall accuracy of the tool irrespective of species.

Result and discussion section is poorly written without sufficient corroboration with earlier works.

Future prospective of the study involving the utility of predicted miRNAs has to be discussed .

Minor comments

In abstract,"..., a pre-trained ....might be miRNAs" . rephrase the sentence.

Caption of Figure 1 is more descriptive, make it short and concise.

First three paragraphs of "Compiling Dataset section" is much descriptive and the second para need to be rephrased.

Figure 2, Step E mention the criteria to designate unfavorable structures.

Point 2 of 10 fold cross validation should be rephrased.

Reviewer #3: This manuscript by Ataei et al. presents a novel tool named “AmiR-P3” is a user-friendly ab initio plant miRNA prediction pipeline. The strength of this pipeline is its user-based customization and its independency or flexibility about the availability of the sequence data. The study offers an alternative novel AI based pipeline to fasten and ease the miRNA research. This pipeline is an excellent tool for miRNA identification in orphan or neglected plant species for which no sequence data is available.

Although, the analysis presented in this study are sufficient and significant but I still recommend to include a table comparing the miRNAS identification stats of AmiRP3 with previously available and widely used tools like miRprefer and miRFINDER .

6. PLOS authors have the option to publish the peer review history of their article (what does this mean?). If published, this will include your full peer review and any attached files.

Reviewer #1: No

Reviewer #2: No

Reviewer #3: **Yes: **Sombir Rao

---

## [Author Response · Author response to Decision Letter 0]

6 Jan 2024

Dear Reviewers,

Thank you for your time and effort in reviewing my manuscript titled “AmiR-P3: An AI-based microRNA prediction pipeline in plants” for publication in PLoS ONE. I appreciate your thoughtful comments and suggestions, which have helped me to improve the manuscript significantly.

In this letter, I will respond to each of your comments in detail. I have also made the requested changes to the manuscript, and these changes are shown with tracking in the Word file. Please feel free to review the tracked changes and let me know if you have any further questions.

I am confident that the revised manuscript is now in a much stronger position for publication. I thank you again for your helpful feedback, and I look forward to hearing from you soon.

Sincerely yours,

Sobhan Ataei

Reviewer #1

Comment #1: Authors stated that, to train a classifier to identify miRNAs, 2 datasets were used, one of known miRNAs and the other of miRNA-resembling sequences that are not miRNAs. For known miRNAs authors used miRBase, but did not explain from where they collected the transcripts resembled with pri-miRNA structures having overlaps with known coding sequences. Specify and explain about the negative dataset.

Answer: We would like to thank the reviewer for this comment. In the current study, we did not use transcripts to create the negative dataset for our classifier. Instead, we aligned the real mature microRNAs of each species to its genome, and then selected genomic windows around the hits with low (5≤LD≤6) but significant (E-value ≤ 0.001) similarity to real mature microRNAs. To further ensure that the selected genomic windows were not microRNAs merely by chance, since microRNAs are generally believed to be non-coding, we chose only coding sequences to create the negative dataset.

To improve the clarity of this point, we changed the text as follows (lines 206 to 211 of the revised manuscript): To create the negative (decoy) datasets, we extracted genomic windows from the genome of each of the species with a minimal similarity to real miRNAs as explained blow:

For each of the nine plants (Table 1) mature miRNA sequences were individually downloaded from miRBase and aligned against the plant genome using BLASTn. For each pair of the aligned sequences, the level of dissimilarity (LD) is computed as: 

LD = Q – Qa + G + M

Comment #2: Authors must have to explain about the significance of taking LD value between 5 and 6.

Answer: We agree with the reviewer that this issue should be clarified. In homology-based identification of microRNAs, to the best of our knowledge, up to 4 (Rajakani et al., 2021, DOI: 10.1007/s12010-021-03500-4) mismatches are allowed in the alignment to accept the sequence as a miRNA candidate. This means that a sequence with up to 4 mismatches to a real miRNA might still be predicted as a functional miRNA. In addition to having enough dissimilarity to known microRNAs, decoy sequences should also have significant similarity to real microRNAs. In this study, to select sequences that are not functional microRNAs, we set the minimum required difference between a query (real miRNA) and the subject (the aligned sequence on the genome) to 5 (LD ≥ 5) and since all alignments with an LD greater than 6 had large E-values (greater than 0.001) and were not considered significant alignments, we set LD=6 as the upper limit for the level of dissimilarity.

Corrected text (lines 213 to 217 of the revised manuscript): It is possible for alignments with LD<5 to be functional miRNAs [53] and alignments with LD>6 often resulted in E-values greater than 0.001, which is considered unacceptable and could not be considered as “significant”. Therefore, only significant alignments (E-value ≤ 0.001) with 5≤LD≤6 were selected as the potential hits.

Comment #3: The authors used DIAMOND software for alignment, though they are using BLAST, explain the usage priority of DIAMOND software.

Answer: Both DIAMOND and BLASTx can be used for nucleotide to protein sequence alignment. However, DIAMOND is much faster than BLASTx (up to 20,000 times faster for short sequences). 

Corrected text (lines 156 to 159 of the revised manuscript): Then, we eliminated the redundant sequences using CD-HIT [50] and chose those sequences that have no overlap with known protein-coding sequences using DIAMOND [40, 41]. In this step, DIAMOND was chosen over BLASTx, as it has a comparable sensitivity, while it is reportedly much faster. 

Comment #4: Whether authors checked for tRNA or rRNA folds before qualifying non-coding sequences as pre-miRNAs predicted structures for positive dataset?

Answer: Thank you for your thoughtful question about whether we have checked for tRNA or rRNA folds before qualifying non-coding sequences as pre-miRNAs predicted structures for positive datasets. We can confirm that we have checked for tRNA or rRNA folds in that analysis. In addition, we also added an additional step to the pipeline in which the pipeline aligns the candidate sequences for predicted miRNAs with rRNA and tRNA sequences from Rfam. This step helps to ensure that any sequences that are identified as pre-miRNAs are not actually tRNAs or rRNAs. I appreciate you bringing this important issue to our attention. Based on this suggestion Fig. 4 was corrected and an additional step was added to the pipeline. we believe that this additional step will help to improve the accuracy of final results and to reduce the number of false positives. We corrected the text as follows:

Corrected text #1 (lines 159 to 164 of the revised manuscript): To obtain rRNA and tRNA sequences, we used the Rfam database (release 14.9) [51]. Then, each of the sequences in the positive dataset was aligned against all tRNAs and rRNAs of the Rfam-derived dataset to ensure that the positive sequences are not rRNA or tRNA. Consequently, for each sequence in each of the compiled datasets, all of the optimal secondary structure(s) of its presumed RNA transcript were predicted by Mfold [35]…

Corrected text #2 (lines 317 to 322 of the revised manuscript): Consequently, around each BLASTn hit, the pipeline selects a wide genomic window, and then aligns it against the sequences in the comprehensive protein dataset (NR). This step is to ensure that each hit is not part of a coding sequence. It, then, aligns all of the remaining sequences with each sequence in the Rfam-derived dataset to remove any possible rRNA and tRNA. Finally, the secondary structure of the presumed RNA sequence is predicted.

Comment #5: The authors mentioned that they have allowed six mismatched positions in the miRNA/miRNA* duplex, and mainly allowed three at asymmetric bulges. But it is assumed that the mismatches at position 9-11 will lead to non-functional miRNA. Whether authors checked the mismatch positions properly?

Answer: It is correct that mismatches in positions 9-11 of a miRNA/miRNA* duplex can lead to a non-functional miRNA. However, this is already implemented in our pipeline. More precisely, we have defined a “region of interest” (ROI) in the miRNA/miRNA* duplex, that is, positions 9-11 by default. Furthermore, we allow up to 6 mismatches in the entire duplex (by default), but we also allow the user to tune the pipeline to reject those structures that contain any mismatches in the ROI. Consequently, we changed the text as follows:

Corrected text (lines 304 to 306 of the revised manuscript): In addition to the above-mentioned criteria for the total number of acceptable mismatches in a miRNA/miRNA* duplex, the user can define new restrictions to further eliminate duplexes that contain any mismatches in specific positions, e. g., positions 9-11.

Comment #6: Why authors compared AmiR-P3 only with PlantMirP2? Why not with other miRNA prediction tools?

Answer: We agree that comparing AmiR-P3 with more miRNA prediction tools can enrich our results and show the relevance of AmiR-P3. We additionally compared the performance of AmiR-P3 with MiRFinder. The results of the comparison are now shown in Fig. 7. Our results suggest that AmiR-P3, in addition to PlantMirP2, outperforms MiRFinder. The results of this comparison are now included in the main text of the article. 

Corrected text #1 (lines 377 to 392 of the revised manuscript): MiRFinder (58) is a pioneer miRNA prediction tool that is accurate and efficient in predicting pre-miRNA sequences from genomic DNA. MiRFinder works in two steps. First, it scans the genome for potential pre-miRNA hairpin sequences. Once potential pre-miRNA hairpin sequences have been identified, it uses a support vector machine (SVM) classifier to classify them as either miRNA or non-miRNA hairpins. The SVM classifier uses some structural features to distinguish between miRNAs and non-miRNA hairpin sequences. PlantMirP2 is another plant miRNA prediction tool that uses a set of energy-based features to train an SVM model for detecting miRNAs with high accuracy. Despite differences in feature extraction and miRNA prediction strategies, these two miRNA prediction tools (with a few minor modifications) can be evaluated using the benchmark datasets established for evaluating AmiR-P3.

To evaluate the AmiR-P3 pipeline, in each iteration, the positive and negative datasets of eight plants were used to train the pipeline, while the correctness of the predictions was evaluated on the remaining plant. For MiRFinder and PlantMirP2, predictions were made by software tools with their default parameter settings. Based on these predictions, the correctness measures were computed individually for each of the nine plants.

Corrected text #2 (lines 461 to 467 of the revised manuscript): In the next step, to make a comparative evaluation of our pipeline, we used nine pairs of positive and negative datasets consisting of predicted RNA secondary structures from real pre-miRNAs and decoy sequences (S6 File) to the classification models of MiRFinder, PlantMirP2, and AmiR-P3. Then we calculated the evaluation metrics for the three programs. As presented in Fig 7, AmiR-P3 outperformed MiRFinder in accuracy, precision, sensitivity, F1 score, and MCC. It also exceeded PlantMirP2 in accuracy, precision, specificity, F1 score, and MCC.

Corrected text #3 (lines 469 to 471 of the revised manuscript): Fig 7. Comparison of evaluation metrics for the classification models of AmiR-P3, MiRFinder and PlantMirP2. Results indicate that AmiR-P3 outperformed MiRFinder in accuracy, precision, sensitivity, F1 score, and MCC. It also exceeded PlantMirP2 in accuracy, precision, specificity, F1 score, and MCC

Reviewer #2

Comment #1: Mention the criteria of eliminating unacceptable in the main text.

Answer: As you suggested, we added the criteria of eliminating unacceptable structures in the main text.

Corrected text (lines 191 to 195 of the revised manuscript): Finally, (E) after feature extraction by CTAnalyzer and the elimination of unacceptable structures (as explained in details in S2 File, structures in which the hit region is not involved in a double-stranded stem, has only a few residues in complementarity, lacks a continuous complementary region, contains inner branches, is complementary to a branched region, or is not located entirely in the same side of the duplex), (F) the positive feature dataset for plant X was compiled.

Comment #2: Have you observed any pattern of nucleotides in the flanking sequences of +ve and -ve miRNAs. Present a sequence logo to show type of nucleotide conservation in the miRNA flanking sequences.

Answer: We greatly appreciate your thorough review and your dedication to enhancing the quality of our work. In response to your question about nucleotide patterns in the flanking sequences of positive (+ve) and negative (-ve) miRNAs, we have performed the analysis as you suggested. Briefly, our investigation did not reveal any observable conservation or significant discrepancies in patterns between the two categories, neither upon conducting a comprehensive analysis of the flanking sequences surrounding the positive miRNAs, nor the negative miRNAs. The absence of apparent conserved patterns further supports that our positive and negative miRNA datasets are unbiased. This observation highlights the robustness of our data selection and reinforces the reliability of our findings. 

For your convenience and for the benefit of interested readers, we have included a comprehensive account of our analysis and its outcomes in Supplementary File S3 and also added following texts to the manuscript: 

Corrected text #1 (lines 246 to 249 of the revised manuscript): In order to examine the potential bias resulting from the occurrence of nucleotide patterns in the flanking sequences of genuine and decoy pre-miRNAs within the positive and negative datasets, we employed WebLogo (version 3.7.12) (54) to create sequence logos. 

Corrected text #2 (lines 408 to 414 of the revised manuscript): 

Checking the compiled datasets for sequence bias

To assess bias caused by nucleotide patterns in flanking sequences of real and decoy pre-miRNAs in both positive and negative datasets, we generate sequence logos using WebLogo. The generated logos, available in S3 File, illustrate the 20 nucleotides both upstream and downstream of each genuine and decoy pre-miRNA in both the positive and negative datasets demonstrate that there is no distinct nucleotide conservation evident in either of the datasets.

Comment #3: if you have taken all the validated miRNAs, then how their structures can be unfavorable?

Answer: we would like to thank the reviewer for this insightful comment. While it is not possible to rule out that some pri-miRNAs have a non-canonical secondary structure, there are other possibilities to explain the observation of unfavorable structures for validated miRNAs.

Firstly, not all miRNAs in miRBase are reliable. miRBase assigns confidence levels to its entries using deep sequencing datasets (Kozomara, A. and S. Griffiths-Jones, 2014, DOI: https://doi.org/10.1093/nar/gkt118), based on mapped read patterns indicating miRNA existence and expression. The high-confidence subset includes miRNAs with consistent, reproducible mapping patterns across samples and experiments, which are more likely to be experimentally validated and functionally relevant (Kozomara, A. et al., 2019, DOI: https://doi.org/10.1093/nar/gky1141). In miRBase (ver. 22.1), only a small portion of plant miRNAs are classified as confident. Consequently, we could not exclusively use confident miRNAs to train the deep learning model. Instead, for each positive dataset, we chose to utilize all reported miRNAs, excluding those with unfavorable secondary structures and overlaps with protein coding sequences to reduce false positive rate.

Secondly, there are at least two reasons that may result in unfavorable “predicted” structures for genuine miRNAs. One possibility is that the 20nt flanks that were added to each pre-miRNA sequence to mimic the pri-miRNAs, may make the resulting sequences longer or shorter than

in vivo pri-miRNAs. This could influence the pri-miRNA folding and make it unfavorable. In addition, even if the presumed pri-RNAs have the correct length, the secondary structure prediction tools can only make predictions based on the sequence of the RNA which are based on mathematical and thermodynamic models and are not necessarily correct. The reason is that such tools cannot account for the effects of environmental factors, such as the attachment of RNA-binding proteins and other molecules, which are known to be able to influence RNA folding. As a result of these limitations, it is possible that some of the validated miRNAs in the positive datasets may have “unfavorable” predicted structures. 

For the above-mentioned reasons, not all miRNAs in miRBase are of the favorable structures. This issue is already considered in the pipeline. Our strategy was to select the positive dataset stringently such that all the sequences in this dataset are checked for relevance of the structural features of a typical plant miRNA. To further highlight this point, in the revised manuscript, we corrected the text as follows.

Corrected text (lines 163 to 180 of the revised manuscript): Consequently, for each sequence in each of the compiled datasets, all of the optimal secondary structures o

---

## [Decision Letter · Decision Letter 1]

30 Jan 2024

PONE-D-23-17993R1AmiR-P3: An AI-based microRNA prediction pipeline in plantsPLOS ONE

Dear Dr. Ataei,

Thank you for submitting your manuscript to PLOS ONE. After careful consideration, we feel that it has merit but does not fully meet PLOS ONE’s publication criteria as it currently stands. Therefore, we invite you to submit a revised version of the manuscript that addresses the points raised during the review process.

After reviewing the assessments and suggestions provided by the reviewers, it is recommended that the authors make minor revisions. The authors should incorporate the suggestions provided by Reviewer 2.

We look forward to receiving your revised manuscript.

Kind regards,

Vibhav Gautam

Academic Editor

PLOS ONE

Journal Requirements:

Additional Editor Comments:

After reviewing the assessments and suggestions provided by the reviewers, it is recommended that the authors make minor revisions. The authors should incorporate the suggestions provided by Reviewer 2.

Reviewers' comments:

Reviewer's Responses to Questions

**Comments to the Author**

1. If the authors have adequately addressed your comments raised in a previous round of review and you feel that this manuscript is now acceptable for publication, you may indicate that here to bypass the “Comments to the Author” section, enter your conflict of interest statement in the “Confidential to Editor” section, and submit your "Accept" recommendation.

Reviewer #1: (No Response)

Reviewer #2: (No Response)

2. Is the manuscript technically sound, and do the data support the conclusions?

Reviewer #1: (No Response)

Reviewer #2: Yes

3. Has the statistical analysis been performed appropriately and rigorously? 

Reviewer #1: (No Response)

Reviewer #2: Yes

4. Have the authors made all data underlying the findings in their manuscript fully available?

Reviewer #1: (No Response)

Reviewer #2: No

5. Is the manuscript presented in an intelligible fashion and written in standard English?

Reviewer #1: (No Response)

Reviewer #2: Yes

6. Review Comments to the Author

Reviewer #1: (No Response)

Reviewer #2: Though authors have addressed majority of my comments, still the following modifications are required.

Authors should at least present the correlation between features as figure or table or heatmap to guide.

Overview of available tool should come in introduction and the lacuna the authors are addressing should follow intermediately after that.

Author must make there training data available for reproducible research.

I am unable to access https://hub.docker.com/r/micrornaproject/amir-p3, authors must verify that the link is working.

English and grammar should be checked thoroughly.

7. PLOS authors have the option to publish the peer review history of their article (what does this mean?). If published, this will include your full peer review and any attached files.

Reviewer #1: No

Reviewer #2: No

---

## [Author Response · Author response to Decision Letter 1]

5 Jul 2024

Dear Reviewers,

Thank you for dedicating your valuable time and expertise to evaluate my manuscript entitled "AmiR-P3: An AI-based microRNA prediction pipeline in plants" for publication in PLoS ONE. I am grateful for your insightful feedback and suggestions, which have greatly contributed to enhancing the quality of the manuscript.

I have carefully considered each of your comments and have diligently incorporated the requested revisions into the manuscript. These modifications have been clearly marked with tracking in the Word document accompanying this letter. Please review these changes at your convenience and do not hesitate to reach out if you require further clarification.

I am confident that the revised manuscript now meets the high standards expected for publication. Your constructive input has been invaluable in this process, and I am eager to receive your final assessment. Thank you once again for your assistance, and I eagerly anticipate your response.

Best regardes,

Sobhan Ataei

Reviewer #2

Comment #1: Authors should at least present the correlation between features as figure or table or heatmap to guide.

Answer: Thank you for your valuable feedback and suggestions on our manuscript “AmiR-P3: An AI-based microRNA prediction pipeline in plants". We appreciate your thorough review and the opportunity to improve our work. Regarding your suggestion to present the correlation between features used in the training process, we have carefully considered your comment and have incorporated a heatmap illustrating these correlations. This heatmap can be found in Supplementary File S7, which we have newly included with the revised submission. We would like to note that Supplementary File S7 is referenced in both line 608 and line 803 of the revised manuscript. We believe that this addition provides a clear and informative visualization to guide readers through the relationships between the features analyzed in our study.

The added text (lines 608 to 609 of the revised manuscript):

The correlation between the features used in the training process is presented in S7 File as a heatmap.

Comment #2: Overview of available tool should come in introduction and the lacuna the authors are addressing should follow intermediately after that.

Answer: Thank you for your valuable suggestion to provide an overview of available tools related to the topic at the beginning of the introduction. We have now included this overview (lines 85 to 100 in the revised manuscript) to better contextualize the research and highlight the existing tools in the field. Limits of currently available software tools are discussed in lines 101 to 140, and advantages of the introduced miRNA prediction tool are briefly addressed in lines 141 to 148. Your feedback has been instrumental in improving the clarity and structure of the manuscript.

The added text (lines 85 to 102 of the revised manuscript):

In the field of plant microRNA prediction, several computational tools have been developed to aid researchers in identifying and characterizing these small regulatory molecules. Tools such as miRNAFinder [32], PlantMirP2 [33], mirMachine [34], PmiRDiscVali [35], miRDeep-P2 [36], SUmir [37], and miRNA Digger [38] have been instrumental in advancing our understanding of plant microRNAs. miRNAFinder utilizes machine learning algorithms to predict novel microRNAs from small RNA sequencing data. PlantMirP2 is a comprehensive tool that integrates multiple prediction algorithms to identify plant microRNAs and their targets. mirMachine employs a homology-based approach for accurate microRNA prediction. PmiRDiscVali focuses on validating predicted plant microRNAs through experimental data analysis. miRDeep-P2 is a widely used tool for discovering known and novel microRNAs from deep sequencing data. SUmir offers a user-friendly interface for predicting microRNAs and their targets in plants. Finally, miRNA Digger is a versatile tool that allows for the identification of conserved and species-specific microRNAs in plant genomes. These tools play a crucial role in expanding our knowledge of plant microRNAs and their regulatory functions, providing valuable insights into the intricate networks governing plant development and stress responses.

Although there are numerous plant miRNA prediction tools available, the majority of them exhibit limitations in terms of flexibility across three critical dimensions:

Comment #3: Author must make there training data available for reproducible research.

I am unable to access https://hub.docker.com/r/micrornaproject/amir-p3, authors must verify that the link is working.

Answer: We appreciate your interest in ensuring the transparency and replicability of our work. We would like to address your concern by confirming that all the necessary data, including the training datasets, required to reproduce the research findings are readily available on our GitHub repository at "https://github.com/Ilia-Abolhasani/amir-p3". Additionally, the software developed during the course of this study, named "AmiR-P3" has been made accessible to researchers through the Docker platform at "https://hub.docker.com/r/micrornaproject/amir-p3". We believe that providing access to both the training data and the developed plant microRNA prediction software tool developed in our research will facilitate the replication and validation of our results by the scientific community. The link to the developed software tool and the training datasets are now available in the revised manuscript (line 44 and line 46 respectively).

The corrected text #1 (lines 41 to 46 of the revised manuscript):

AmiR-P3 is provided as a docker container, which is a portable and self-contained software package that can be readily installed and run on any platform, and is freely available for non-commercial use from: 

https://hub.docker.com/r/micrornaproject/amir-p3

The source code of AmiR-P3 is also freely available from:

https://github.com/Ilia-Abolhasani/amir-p3

The corrected text #2 (lines 356 to 361 of the revised manuscript):

For simplicity, we provide AmiR-P3 as a Docker container (freely available from https://hub.docker.com/r/micrornaproject/amir-p3), allowing the users to install the pipeline in a single step without the need to pre-install any other utilities. The source code of AmiR-P3 and all the other data neccesarry to make the research reproducible (including the training datasets) are freely available from https://github.com/Ilia-Abolhasani/amir-p3 .

Comment #4: English and grammar should be checked thoroughly.

Answer: We appreciate your comments and have carefully reviewed the manuscript to address the English and grammar issues. We have made revisions in line 37, 59, 83, 116, 118, 121, 131, 177, 188, 195, 212, 232, 235, 275, 307, 311, 350, 358, 432, 462, 516, 522, 530, 550, and 553 (in the revised manuscript) to ensure the manuscript meets high linguistic standards. The track changes version of the article reflects these corrections, ensuring clarity and correctness throughout the text. We hope that the revised manuscript now meets your expectations regarding language quality.

---

## [Editor Report · Decision Letter 2]

16 Jul 2024

AmiR-P3: An AI-based microRNA prediction pipeline in plants

PONE-D-23-17993R2

Dear Dr. Ataei,

We’re pleased to inform you that your manuscript has been judged scientifically suitable for publication and will be formally accepted for publication once it meets all outstanding technical requirements.

Kind regards,

Vibhav Gautam

Academic Editor

PLOS ONE

Additional Editor Comments (optional):

Manuscript is recommended for publication.
---

## [Editor Report · Acceptance letter]

24 Jul 2024

PONE-D-23-17993R2 

PLOS ONE

Dear Dr. Ataei, 

I'm pleased to inform you that your manuscript has been deemed suitable for publication in PLOS ONE. Congratulations! Your manuscript is now being handed over to our production team.

Kind regards, 

on behalf of

Dr. Vibhav Gautam 

Academic Editor

PLOS ONE